# Job Satisfaction within the Grassroots Healthcare System in Vietnam’s Key Industrial Region—Binh Duong Province: Validating the Vietnamese Version of the Minnesota Satisfaction Questionnaire Scale

**DOI:** 10.3390/healthcare12040432

**Published:** 2024-02-08

**Authors:** Chin Minh Huynh, Chuong Hong Nguyen, Khoa Nguyen Dang Le, Phuong Thi Ngoc Tran, Phuong Minh Nguyen

**Affiliations:** 1Binh Duong Province’s Department of Health, People’s Committee of Binh Duong Province, Thu Dau Mot City 750000, Vietnam or soyte@binhduong.gov.vn (C.M.H.); hongchuongdr@gmail.com (C.H.N.); lekhoa0097@gmail.com (K.N.D.L.); 2Department of Pediatrics, Faculty of Medicine, Can Tho University of Medicine and Pharmacy, Can Tho City 900000, Vietnam; 2253010674@student.ctump.edu.vn

**Keywords:** job satisfaction, MSQ short form, questionnaire value, grassroots healthcare worker, Vietnam healthcare

## Abstract

***Background***: The grassroots healthcare system is the closest and most community-oriented force, working as an extended arm of the primary healthcare network to implement healthcare programs at the household level. Its comprehensive development is a crucial task set by the Vietnamese government. Job satisfaction significantly influences the performance of healthcare staff within this system. ***Objective***: to assess job satisfaction among healthcare staff using the short-form of the Minnesota Satisfaction Questionnaire while also evaluating the Vietnamese translation of this scale. ***Methods***: A descriptive cross-sectional study with analysis based on the responses of 587 healthcare staff using the Vietnamese-translated version of the MSQ short-form scale. The response data from the participants were subjected to CFA, and if the proposed CFA model did not fit the data, EFA was conducted. ***Results***: The results indicate that the new model, which evaluates job-related factors in three distinct groups, is more suitable than the original model. The 14 questions of the MSQ scale were analyzed and categorized into Autonomy, Obligation, and Specificity based on the participants’ responses. The confirmatory factor analysis (CFA) conducted on the new model demonstrated favorable fit indices: CFI = 0.934, TLI = 0.917, GFI = 0.919, and RMSEA = 0.093 (90% CI: 0.085–0.102). ***Conclusions***: The Vietnamese version of the MSQ short form demonstrates reliability and validity. It also provides additional data on the effectiveness of the MSQ short form in measuring job satisfaction.

## 1. Introduction

In Vietnam, the grassroots healthcare system is an essential component of the healthcare system. It offers first-line medical help for the population of over 97 million, educates the population to improve public health, maintains a disease-free environment, and prevents disease outbreaks. To fulfill these responsibilities, the grassroots healthcare system relies on grassroots healthcare workers who carry out their duties regularly without a fixed salary [1]. However, a recent evaluation in Vietnam showed that the performance of these healthcare workers is unsatisfactory [2,3,4,5,6]. One of the most influential factors that affect their performance is job satisfaction.

According to Kebriaei and Moteghedi, job satisfaction refers to the subjective experience of individuals regarding their jobs and the various associated components. It represents the level of alignment or disparity between an employee’s values and the actual provisions within their working environment [7]. The World Health Organization predicted that approximately 47% of Southeast Asian healthcare professionals (nurses, midwives, and doctors) would resign due to job dissatisfaction [8]. High job satisfaction has also been linked to improved patients’ health outcomes and decreased mortality rates. Currently, multiple job satisfaction scales are used worldwide, such as the Job Satisfaction Survey (JSS), Job Descriptive Index (JDI), and the Andrew and Withey Job Satisfaction Questionnaire. However, the reliability of these scales in evaluating factors affecting job satisfaction is low.

In 1967, Weiss developed the Minnesota Satisfaction Questionnaires short form (MSQ short form) [9,10]. The questionnaires focus on extrinsic and intrinsic factors affecting job satisfaction [11,12,13]. The MSQ short form initially contained 20 items evaluated on a 5-point Likert-type response scale ranging from very dissatisfied (1) to very satisfied (5). The MSQ is stable with a very good Alpha value (0.85 to 0.91). Therefore, it has been translated into many languages and used in many countries worldwide [14,15]. The self-report Minnesota Satisfaction Questionnaires are employed in healthcare settings to assess various aspects of job satisfaction. These studies revealed that decreased job satisfaction significantly contributes to work reversals, especially in healthcare services. Additionally, in Cyprus, the confirmatory factor analysis (CFA) results for the three-factor model of the Gr-MSQ-short, as proposed by Weiss, yielded notable findings.

Although there is no standardized tool for measurement, a multilevel study conducted by a group of Vietnamese authors in 2020 also revealed that work-related challenges and seniority significantly impact job satisfaction among healthcare workers. In addition, based on data from The Vietnam Health Statistic Yearbook in 2015, this study suggested that individual factors such as gender, age, and educational level are associated with job satisfaction. The factors related to healthcare facilities and management from the supervising entity also influence job satisfaction among healthcare personnel. However, as Weiss discussed in the Minnesota Satisfaction Model, other objective factors have not received much attention.

Currently, in Vietnam, no standardized tool is used to measure job satisfaction among grassroots healthcare workers. Instead, provincial health departments regularly conduct surveys to assess the satisfaction of healthcare personnel working under their management. However, these surveys lack synchronization, and the survey models vary across different locations, resulting in numerous inconsistencies and shortcomings. The lack of a common survey tool or a unified guideline from the Vietnamese Ministry of Health to assess the satisfaction of healthcare personnel contributes to these issues. Therefore, we conducted this study to translate and evaluate the Vietnamese version of the MSQ short form, a questionnaire used to measure the job satisfaction of grassroots health workers. The primary objective was to assess the job satisfaction of healthcare personnel using the MSQ assessment tool. Additionally, we aimed to analyze the factors of the Vietnamese version of MSQ to create an improved version, serving as a pioneering and widely used tool for evaluating job satisfaction among healthcare personnel in Vietnam. This annual evaluation activity is carried out monthly or quarterly by various levels of healthcare management in Vietnam to enhance the overall well-being of healthcare personnel.

## 2. Materials and Methods

### 2.1. Study Population and Research Tools

A cross-sectional study was conducted on all healthcare staff within the village-hamlet-sub-hamlet healthcare system (the grassroots healthcare system) in Binh Duong province, Vietnam, from February 2023 to October 2023. Binh Duong is a province in southern Vietnam, ranking fifth in terms of population size in the country. The province comprises 596 village–hamlet–sub-hamlet areas, corresponding to 596 healthcare staff within this system. The study initially aimed to include the entire population; however, after excluding healthcare staff who did not fully participate in the research process, those who had not completed one year of service, and those who declined to participate, the final sample consisted of 587 healthcare staff.

#### The Minnesota Satisfaction Questionnaires—Short Form

To ensure the suitability and accessibility of the questionnaire for the study participants, the original English version by Weiss (1967) was translated into Vietnamese. The Vietnamese translation version is based on the methodology of a previous study on score-based translation applied in the field of clinical medicine [16]. The Vietnamese translation was conducted by a Vietnamese native who holds a Master’s degree in English language. To increase the reliability, the translated version was back-translated into English by another individual who is also a Master’s degree holder in English and was unfamiliar with the original English version. The evaluation between the original English version and the back-translated version was conducted using independent comparisons by two language experts with Master’s degrees in English language and culture. The evaluation criteria included semantic equivalence, idiomatic equivalence, experiential equivalence, and conceptual equivalence, with a rating scale of 1 point for equivalence and 0 for non-equivalence. Questions that did not achieve absolute equivalence (10/10) across the four criteria and proposed adjustments to the translated MSQ content (suggested by English language experts and a Vietnamese specialist in healthcare management) underwent a second evaluation for equivalence based on the four aforementioned criteria. The final version of the MSQ was then created and utilized for pilot testing. Subsequently, a group of volunteers was given both the original English version and the translated version for trial use. After evaluating the accuracy and reliability of the translation, the new Vietnamese short-form version of the MSQ was deemed suitable for evaluating the satisfaction of the healthcare staff participants.

Additionally, we reviewed and categorized this scale into specific factor groups based on the analysis of the 20 questions included in the scale. For each question, participants rated their level of satisfaction using a Likert scale with 5 levels: (1) very unsatisfied, (2) unsatisfied, (3) satisfied, (4) very satisfied, and (5) extremely satisfied. The results were calculated by summing the scores of the 20 items or calculating the average score for each analyzed factor. Higher scores indicate higher levels of job satisfaction. 

### 2.2. Data Collection and Analysis Methods

The study participants self-completed their evaluations using the pre-printed Vietnamese version of the Minnesota Satisfaction Questionnaire on paper. They then deposited them into designated collection boxes at all primary healthcare stations throughout the Binh Duong province. Subsequently, the research team collected these responses, ensuring participant confidentiality and privacy. The obtained results were encoded and entered into the Edidata 3.1 software to create a database for analysis in the study.

The collected data were subjected to exploratory factor analysis (EFA) using SPSS version 25.0 software (IBM Corp., New York, NY, USA) and confirmatory factor analysis (CFA) using AMOS version 25.0 software.

#### 2.2.1. The Structural Validity of the MSQ Scale Will Be Assessed via Factor Analysis

Confirmatory Factor Analysis (CFA): Evaluate using indices including Comparative Fit Index (CFI), Tucker–Lewis Index (TLI), Goodness-of-Fit Index (GFI), and Root Mean Square Error Approximation (RMSEA) to determine the model fit. The model is appropriate when CFI, TLI, GFI ≥ 0.9, and RMSEA ≤ 0.08. If the proposed CFA model does not fit the data well, exploratory factor analysis (EFA) will be conducted.

Exploratory Factor Analysis (EFA): Utilize Principal Component Analysis (PCA) with Promax rotation; factor loading ≥ 0.5 will be employed. The appropriateness of EFA will be determined based on the total extracted variance, Kaiser–Meyer–Olkin (KMO) test, and Bartlett’s test of sphericity. EFA is considered suitable when the total extracted variance is above 50%, 0.5 ≤ KMO ≤ 1, and Bartlett’s test is statistically significant (Sig. < 0.05).

#### 2.2.2. Reliability

The internal consistency will be estimated using Cronbach’s Alpha coefficient. The scale is considered to have internal consistency when Cronbach’s Alpha value > 0.5.

### 2.3. Ethical Approval

This study was approved by the Department of Health of Binh Duong Province and the Ethic Committee of Can Tho University of Medicines and Pharmacy (Approval number: 22.005.NCS/PCT-HDDD).

## 3. Results

### 3.1. Translation and Cross-Cultural Adaptation of the MSQ Short Form

Following the translation and cross-cultural adaptation process, a significant equivalence was achieved between the back-translated into English and English versions. The adjusted questions were re-evaluated for equivalence a second time based on the four established criteria, achieving a level of equivalence ranging from 8/10 to 10/10. The final version of the MSQ was then created and utilized for pilot testing. The Vietnamese version of the MSQ was developed after pilot testing, and necessary adjustments were made based on volunteers’ feedback (Appendix A). 

### 3.2. Demographic Characteristics

Table 1 shows the general characteristics of the study participants. The male-to-female ratio was 1:5.11. The average age was 54.95 ± 10.92. Approximately 54.5% of grassroots healthcare workers had professional qualifications related to the medical field. The majority of them also held additional local responsibilities, with 72.7% serving as population collaborators and 64.2% serving as nutrition collaborators. The average time grassroots healthcare workers dedicated to their jobs was 47.86 ± 41.5 h per month. The majority of them received allowances for these activities (94.9%), with 77.7% recorded to have received more than the minimum amount.

### 3.3. Assessed via Factor Analysis

The assessment of the initial 20-item model yielded the following results: χ^2^ = 2874.410, df = 165, *p* < 0.001, GFI = 0.596, CFI = 0.680, and RMSEA = 0.167. These findings indicated that the model was not suitable for use. Consequently, a new model was constructed based on the scales derived from exploratory factor analysis.

#### 3.3.1. Exploratory Factor Analysis

Factor loadings ranged from 0.57 to 0.912 (Table 2). Factor analysis yielded three factors, explaining 65.766% of the variance. The KMO measure is 0.909 (>0.5), and Bartlett’s test with a *p*-value < 0.001 indicates that the correlations among the items are suitable for PCA.

Items 6, 7, 8, 12, 13, and 19 were excluded from the analysis due to high cross-loadings.

The remaining items were arranged into three groups. The first factor consists of seven items: MSQ10, MSQ9, MSQ11, MSQ20, MSQ18, MSQ5, and MSQ4. This factor explains 48.442% of the variance. It represents the opportunities the job provides for staff to assert themselves, referred to as the “Autonomy” factor. The second factor consists of four items: MSQ15, MSQ16, MSQ17, and MSQ14. This factor accounts for 10.861% of the variance. It represents the rights and benefits the staff gain from their jobs, referred to as the “Obligation” factor. The third factor comprises three items: MSQ1, MSQ3, and MSQ2. This factor accounts for 6.463% of the variance. It represents staff’s perceptions of the nature of their jobs, referred to as the “Specificity” factor. The Cronbach’s Alpha coefficient value of 0.924 indicates a high level of internal consistency among the items (Table 3).

#### 3.3.2. Confirmatory Factor Analysis of the New Model

The re-evaluation of the confirmatory factor analysis (CFA) on the new model revealed that the fit indices for the three-group-factor model (Appendix B), including the covariance between factors (Figure 1), indicated the appropriateness of this model, such as CFI = 0.934, TLI = 0.917, GFI = 0.919, and RMSEA = 0.093 (90%CI: 0.085–0.102).

Based on the factor analysis results presented in Table 2, the study identified three groups of factors that exhibited moderate correlations with each other, ranging from 0.21 to 0.26. The factor loadings ranged from 0.8 to 1.02. Several squared correlation coefficients fell within the range of 0.12 to 0.32, indicating that the factors provided acceptable explanations for the observed variations in the items used in the CFA model.

## 4. Discussion

### 4.1. Principal Findings and Comparisons

In the study, the validity and reliability of the Vietnamese version of the MSQ short form were assessed based on data collected from 587 healthcare staff in Binh Duong province using the questionnaire. The new model was more appropriate than the original one, with three distinct groups of job-related factors being evaluated.

Based on the CFA result, the initial three-group model had insufficient validity. This result is similar to previous studies where the original model was incompatible with the studied regions [13,14]. According to Lakatamitou’s study, the original three-group model has the following values: RMSEA = 0.12 (95%CI: 0.117–0.133), AGFI = 0.889, CFI = 0.897, and TLI = 0.133) [14].

The EFA analysis resulted in a model with 14 items rearranged into three groups, which accounted for 65.766% variances. However, the study of Lakatamitou et al. (2020) resulted in 2 groups with 15 items. The two groups (Supervisor/Autonomy and Task Enrichment) account for 58.0% variances [14]. In Martin’s study (2012), the result is a 2-group model with 10 items; the 2 groups (Supervisor/Empowerment and Task Enrichment) account for 61.185 variances [13].

The first group factor, “Autonomy”, includes four items: MSQ10, MSQ9, MSQ11, and MSQ20. Initially, these items were categorized under “Intrinsic job satisfaction”. However, the revised model found that it was more appropriate to include MSQ18 and MSQ5, which address the relationships between grassroots healthcare workers, supervisors, and co-workers.

The second group factor, labeled “Obligation”, comprises MSQ15, MSQ16, MSQ17, and MSQ14. Lakatamitou et al.’s (2020) research associated these items with “Task Enrichment”. The original model included them in the “Intrinsic Job Satisfaction” category [14]. However, the change to “Obligation” better captures the essence of these items.

The third group factor, “Specificity”, consists of MSQ1, MSQ3, and MSQ2. These items pertain to the ability of grassroots healthcare workers to perform various tasks independently to complete their jobs. Initially, these items were placed in the “Intrinsic Job Satisfaction” category. However, the revised categorization as “Specificity” more accurately reflects their nature.

Items MSQ6, MSQ7, MSQ8, MSQ12, MSQ13, and MSQ19 were excluded due to cross-loading. In Vietnam, grassroots healthcare workers do not undergo regular inspections. Instead, they receive specific tasks based on the annual regulations set by the Ministry of Health. It is also important to note that these workers are considered volunteers and receive allowances according to government regulations, which often do not adequately compensate for their extensive workload.

The MSQ short form demonstrated good reliability in this study, as indicated by a Cronbach’s Alpha value of 0.924. This finding aligns with a previous report by Lakatamitou et al. (2020), which reported a Cronbach’s Alpha value of 0.955 [14]. These results highlight the reliability and internal consistency of the Vietnamese version of the MSQ short form.

Regarding statistical validity, the chi-squared test indicated that the model was unsuitable. However, when considering the GFI, TLI, CFI, and RMSEA values, it was evident that the model was indeed suitable. This discrepancy can be attributed to the large sample size [17]. Overall, we concluded that the three-group model aligns well with the original model and exhibits similarities to previous studies (Table 4). Therefore, we can confidently assert that the 14-item MSQ short form demonstrates good construct validity.

### 4.2. Implications and Strength

Due to the inconsistency in methods used in studying job satisfaction among healthcare workers in Vietnam, we conducted this study to address the need for standardization in measuring job satisfaction, particularly among grassroots healthcare workers. We aimed to translate and evaluate the Vietnamese version of the MSQ short form to establish a standardized approach. Additionally, considering the regional and temporal variations, our study aimed to provide a foundation for future research that identifies factors influencing job satisfaction.

The MSQ short form, known for its reliability, has been translated into multiple languages to measure workers’ job satisfaction. However, our research was the first to translate and assess the reliability of the Vietnamese version of the MSQ short form. With a relatively large sample size for the pilot (587 participants), we ensured a high level of accuracy and minimized the potential for deviation in our findings.

### 4.3. Limitations 

This research was conducted in the year following the decline in the COVID-19 pandemic. During this period, there were various alterations in the composition of the health workforce and changes in factors influencing job satisfaction [15,19]. It is important to note that the study solely concentrated on healthcare workers in one specific province, and therefore, the results cannot be generalized to other regions in Vietnam. Furthermore, due to time constraints, we could not assess the test–retest reliability of the questionnaires.

### 4.4. Further work

Although job satisfaction among grassroots healthcare workers is unlikely to change significantly quickly, especially without modifications to healthcare policies, previous studies have emphasized the correlation between job satisfaction and factors such as seniority and education level. Consequently, further research is warranted to explore how these factors influence the reliability of the MSQ. Since our study focused solely on one province, conducting additional studies with larger sample sizes encompassing Vietnam is crucial.

Our research relied on self-reported responses from grassroots healthcare workers. However, we did not employ any tools to assess the honesty of these responses. Therefore, future studies should incorporate methods to analyze the honesty and accuracy of participants’ answers.

## 5. Conclusions

This study contributes additional data to the global community regarding validating the MSQ short form in effectively assessing job satisfaction levels among healthcare staff. Alongside previous research and after evaluating and implementing necessary adjustments, this study found that the Vietnamese version of the MSQ short form demonstrates the required reliability and validity for measuring job satisfaction. Moreover, the study suggests the need for further in-depth research to evaluate the scale’s psychometric properties in different localities and assess the test–retest reliability of the measurement tool.

## Figures and Tables

**Figure 1 healthcare-12-00432-f001:**
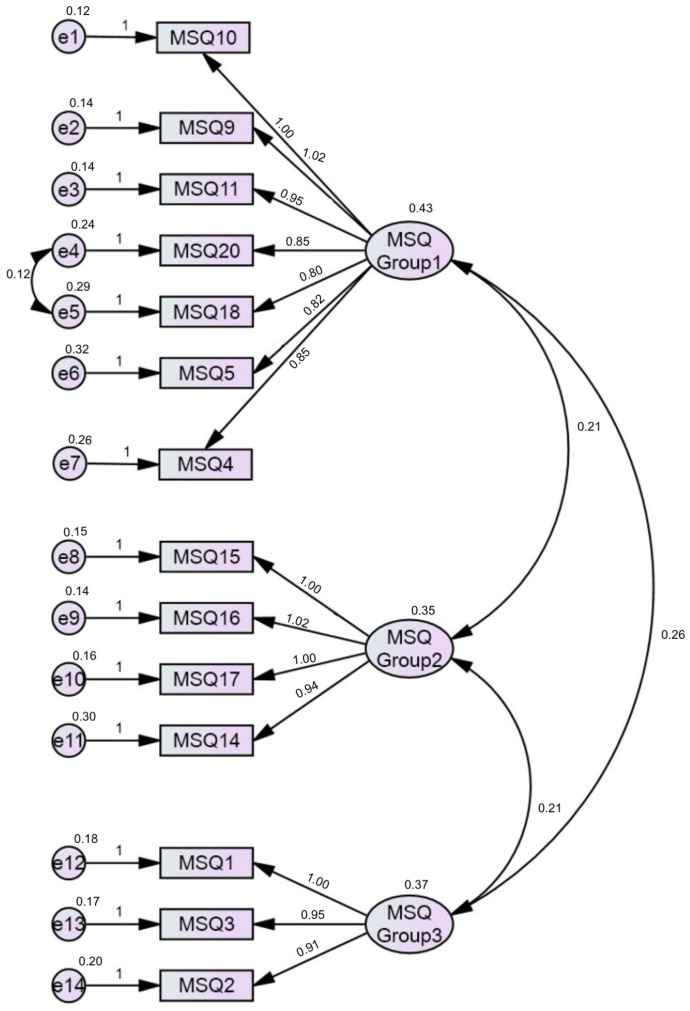
Structure of the MSQ short form.

**Table 1 healthcare-12-00432-t001:** The general characteristics of the study participants (*n* = 587).

Characteristic		Frequency (*n*)	Percentage (%)
Gender	Male	96	16.4
	Female	491	83.6
Age	30 years and younger	12	2.0
	31 to 60 years	373	63.5
	Older than 60 years	202	34.4
	Mean ± Standard Deviation	54.95 ± 10.92	
	Median	58.00	
	Youngest age	21	
	Oldest age	89	
Academic Level	Elementary	25	4.3
	Secondary	289	49.2
	High School	206	35.1
	University or Higher Education	67	11.4
Qualifications	Medicine-related	320	54.5
	Non-medicine-related	267	45.5
Job responsibilities that require multiple positions	Population collaborator	427	72.7
	Nutrition collaborator	377	64.2
	Head of hamlet	18	3.1
	Propagandist of the Vietnam Women’s Union	146	24.9
	Others	116	19.8
Working time (day/month)	Less than 1 day/month	158	26.9
	Between 1 and 4 days/month	382	65.1
	More than 4 days/month	47	8.0
	Mean ± Standard Deviation	47.86 ± 41.5	
	Median	32	
	Least time working	2 (h)	
	Most time working	240 (h)	
Receiving allowances	Yes	557	94.9
	No	30	5.1
The amount of allowances received per month	Less than the minimum	101	17.2
	The minimum and more	456	77.7

**Table 2 healthcare-12-00432-t002:** Exploratory factor analysis of MSQ.

Item	Group		
	1	2	3
MSQ10. The chance to tell people what to do	0.912		
MSQ9. The chance to do things for other people	0.867		
MSQ11. The chance to do something that makes use of my abilities	0.837		
MSQ20. The feeling of accomplishment I get from the job	0.773		
MSQ18. The way my co-workers get along with each other	0.720		
MSQ5. The way my boss handles his/her workers	0.704		
MSQ4. The chance to be “somebody” in the community	0.570		
MSQ15. The freedom to use my own judgment		0.884	
MSQ16. The chance to try my own methods of doing the job		0.856	
MSQ17. The working conditions		0.715	
MSQ14. The chances for advancement on this job		0.659	
MSQ1. Being able to keep busy all the time			0.887
MSQ3. The chance to do different things from time to time			0.726
MSQ2. The chance to work alone on the job			0.715
Percentages of Variances	48.442	10.861	6.463

**Table 3 healthcare-12-00432-t003:** Cronbach’s Alpha value of MSQ.

Item	Mean	Sd.	Corrected Item–Total Correlation	Cronbach’s Alpha If Item Deleted	Cronbach’s Alpha Coefficient
MSQ1. Being able to keep busy all the time	3.2419	0.74087	0.610	0.920	0.924
MSQ2. The chance to work alone on the job	3.3305	0.71058	0.610	0.920	
MSQ3. The chance to do different things from time to time	3.4174	0.70921	0.676	0.918	
MSQ4. The chance to be “somebody” in the community	3.6934	0.75134	0.685	0.918	
MSQ5. The way my boss handles his/her workers	3.7819	0.77733	0.600	0.921	
MSQ9. The chance to do things for other people	3.7104	0.76257	0.758	0.915	
MSQ10. The chance to tell people what to do	3.7376	0.73850	0.741	0.916	
MSQ11. The chance to do something that makes use of my abilities	3.6814	0.72666	0.739	0.916	
MSQ14. The chances for advancement on this job	3.0767	0.78331	0.526	0.923	
MSQ15. The freedom to use my own judgment	3.2470	0.70733	0.574	0.921	
MSQ16. The chance to try my own methods of doing the job	3.2964	0.71362	0.608	0.920	
MSQ17. The working conditions	3.2572	0.71330	0.670	0.918	
MSQ18. The way my co-workers get along with each other	3.7785	0.74725	0.660	0.919	

**Table 4 healthcare-12-00432-t004:** Values in CFA Analysis of the MSQ short form.

Studies	χ^2^	df	GFI	TLI	CFI	RMSEA	90%CI
Original model with 20 items [9]	453.507	167	-	-	0.741	0.111	0.099–0.123
Sousa et al. (2011) model with 11 items [18]	73.653	42	-	-	0.908	0.074	0.045–0.101
Martins (2012) mode with 10 items [13]	52.091	33	-	-	0.969	0.065	0.027–0.097
Lakatamitou (2020) model with 15 items [14]	237.743	81	0.906	0.916	0.935	0.08	0.068–0.091
Our study (2023) with 14 items	466.699	73	0.905	0.917	0.934	0.093	0.085–0.102

## Data Availability

Data are contained within the article. Data sharing does not apply to this article.

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
