# Peer review of "Job Satisfaction within the Grassroots Healthcare System in Vietnam’s Key Industrial Region—Binh Duong Province: Validating the Vietnamese Version of the Minnesota Satisfaction Questionnaire Scale"

_healthcare, 2024, doi:10.3390/healthcare12040432_

Round 1

Reviewer 1 Report

Comments and Suggestions for Authors

1. The method part of this paper is vague and needs to be supplemented with specific content.

2. The introduction part lacks the summary and analysis of the existing literature, so as to highlight its own innovation.

3. Some figures and tables are not clear, it is recommended to modify.

Author Response

Please see the attachment, thank you!

Reviewer 2 Report

Comments and Suggestions for Authors

This article is very interesting.

Author Response

Please see the attachment, thank you!

Reviewer 3 Report

Comments and Suggestions for Authors

The paper titled " Job satisfaction within the grassroots healthcare system in Vi-etnam's key industrial region: Validating Vietnamese Version of MSQ-scale" appears intriguing. However, it does have some significant shortcomings that the authors should address. My comments are provided in numbered form below.

The introduction effectively establishes the importance of grassroots healthcare in Vietnam but fails to adequately contextualize the specific challenges faced by healthcare workers in this sector. Expanding on the unique conditions and obstacles these workers encounter would provide a stronger rationale for the study's focus on job satisfaction within this specific context.While the introduction highlights the issue of unsatisfactory performance among healthcare workers, it lacks a direct link between job satisfaction and performance outcomes. Including evidence or studies that explicitly connect job satisfaction to performance in healthcare settings would strengthen the argument for focusing on this area of research. The section on job satisfaction scales briefly mentions several tools but does not provide enough comparative analysis to justify the choice of the MSQ-short form over others. A more detailed discussion of why this particular scale is most suitable for the Vietnamese grassroots healthcare context would enhance the introduction's persuasiveness. The introduction presents statistical data and predictions from the WHO but does not integrate this information seamlessly into the narrative. A more cohesive presentation of these statistics, linking them directly to the situation in Vietnam, would create a more compelling case for the study. While the MSQ-short form's history and reliability are mentioned, the introduction misses the opportunity to discuss any previous adaptations of the scale in similar contexts or its relevance to the Vietnamese cultural setting. Including such information would demonstrate the scale's applicability and relevance to the study's objectives.

The translation process for the Minnesota Satisfaction Questionnaires (MSQ) short form, although involving back-translation, lacks a thorough validation process. It's crucial to involve subject matter experts or healthcare professionals in the translation validation to ensure the questionnaire's cultural and contextual relevance to Vietnamese healthcare staff. The exclusion criteria for the sample population are briefly mentioned but lack detailed justification. Specifically, the rationale behind excluding healthcare staff with less than one year of service or those not fully participating in the research process is not clearly articulated, potentially leading to sampling bias or an unrepresentative sample. While data collection methods are described, the study does not address potential biases or limitations inherent in self-reported data, especially considering the sensitive nature of job satisfaction. Implementing additional measures to validate responses or including complementary qualitative data could enhance the robustness of the findings. The statistical methods used for analysis, including exploratory and confirmatory factor analysis, are appropriate. I suggest that please follow these reference article for validity and reliability explanation as well as cite purpose (10.3233/WOR-211363). However, the study could be strengthened by providing more detailed information on the process of categorizing the MSQ scale into specific factor groups, ensuring that the categorization process is transparent and replicable.

 The study, concentrated solely on healthcare workers in Binh Duong province, lacks a broader perspective. Expanding the scope to include other regions in Vietnam would enhance the generalizability of the findings and provide a more comprehensive understanding of job satisfaction among grassroots healthcare workers. Conducted post-COVID-19, the study captures a unique moment in healthcare which might not reflect the typical working conditions or satisfaction levels. Future studies should consider a more diverse timeframe to capture a broader spectrum of the healthcare environment. The absence of a test-retest reliability assessment limits the study's strength in establishing the Vietnamese MSQ-short form's consistency over time. This aspect is critical for ensuring the tool's reliability for longitudinal studies or repeated measures. Relying solely on self-reported data without methods to validate the honesty or accuracy of responses may introduce bias. Future studies should incorporate mechanisms to cross-check the authenticity of participant responses to ensure data integrity.

I hope that the provided comments serve to further strengthen the rigor and depth of this paper. They are intended to highlight areas of potential enhancement and are not a critique of the research's foundational merit. It is my sincere hope that the authors perceive these insights in the constructive spirit with which they are shared. Thank you for the opportunity to review this significant piece of work.

Comments on the Quality of English Language

Follow the detailed comments 

Author Response

Please see the attachment, thank you!

Reviewer 4 Report

Comments and Suggestions for Authors

Abstract:

The abstract provides background regarding the grassroots healthcare system in the Vietnam and the objective (to assess job satisfaction using an instrument adapted from the Minnesota Satisfaction Questionnaire short form) as well as evaluating the Vietnamese translation of the instrument. The abstract also describes methods in terms of testing the instrument using responses from 568 healthcare staff. The paper concludes that the translation is more suitable than the original model. Including a few sentences regarding the factor analysis of the responses might improve the abstract.

Introduction:

The introduction describes the “grassroots healthcare system” in Vietnam. The authors note that grassroots healthcare workers carry out their duties regularly without a fixed salary. This arrangement may be a bit strange for some readers so that a bit more description of the training and duties of the participants might be in order. The authors also noted that the performance of healthcare workers is unsatisfactory.  I would like to know in what respect the performance is unsatisfactory.  It would be helpful to describe the volunteer nature of their participation in the abstract and in the introduction. I suppose that job satisfaction in volunteers can be measured just like job satisfaction for people who are paid. Also, in the demographics section of the paper the instrument measures “allowances.” Does this convert the “volunteers” to paid professionals?

The authors provide information relative to defining job satisfaction. They reference the World Healthcare organization study that predicts nearly half of Southeast Asian healthcare professionals intended to resign due to job dissatisfaction.

The introduction also describes the Insulin Satisfaction Questionnaire-short form, a 20-item survey using Likert response skills. This instrument, apparently, has been translated into many languages and used in many countries worldwide.

Materials and methods:

The instrument was tested on “all healthcare staff within the village-Hamlet-sub-Hamlet healthcare system in Binh Duong province between February and October 2023.

The original instrument was translated into Vietnamese. To increase reliability (the authors use the word ensure which may be inappropriate) the translated version was back translated into English to test the reliability of the translation.

The researchers used exploratory factor analysis and confirmatory factor analysis to evaluate the responses.  For the exploratory factor analysis The authors used principal component analysis with promax rotation employing a Kaiser-Meyer- Olkin test and Bartlett’s test of sphericity. It might be helpful to explain why they promax rotation was employed. They also used araw coefficient alpha test for reliability (Prof. Cronbach who developed the test preferred that it not be called Cronbach’s alpha).

Both the factor analysis and the reliability test are well-accepted textbook methods of conducting this analysis.

Results:

The demographic of the respondents was predominantly female (83.6%) 31 to 60 years of age (63.5%) with a median of 58 years. About half of the participants had a secondary education. A surprising number of the participants were “non-medicine -related” (45.5%).  A substantial number of respondents worked between one and four days per month. 94.9% received allowances with many of them receiving the minimum and more.

The factor analysis results prompted the authors to conclude that the model “that is not suitable for use.” Accordingly, they developed the new model based on skills derived from exploratory factor analysis. The exploratory factor analysis found three factors that explain 65.8% of the variance. Six

 items were excluded from the analysis due to high cross-learning. The authors report the factor loadings for the three groups along with the percentage of the variance explained. The authors called the first group “autonomy factor,” the second “rights and benefits that the staff gain” and the third is defined as a “specificity factor.” The raw coefficient alpha of 0.924 indicates a high level of reliability.

The authors then test the new model using confirmatory factor analysis, concluding that the new model is quite appropriate based on the fit indices for the model.

Generally, the authors’ factor analysis is extremely well done, appropriate and conclusive.

Figure 1 should probably be expanded. It is pretty small in the initial draft of the paper. Also, the figure is not discussed or described. A more detailed description of it might add substantial understanding to the description of the results in the ultimate instrument that the authors development.

Discussion:

The authors describe the study as testing the validity and reliability of the Vietnamese version of the instrument. I’m not sure that the study actually assessed validity. It would be important to describe how this study validated the model.  I am not sure that factor analysis is an assessment of validity.  In the discussion section the authors refer to the chi-square test is relating to the validity of the model.

The authors also noted an inconsistency between the chi-square test that indicates that the model was unsuitable and the confirmatory factor analysis that concludes that the model was suitable. They explain the discrepancy by going to the large sample size. I am not sure that the large sample size can explain the discrepancy.

The authors explained that the three-group model aligns well with the original model and exhibits similarities to previous studies so they concluded that the 14-item revision “demonstrates good construct validity.” I am not sure that similarity with the original model and other studies provides the basis for “construct validity.”

Author Response

Please see the attachment, thank you!

Round 2

Reviewer 1 Report

Comments and Suggestions for Authors

The author modified the problem that was first raised, but by modifying the traces, a more serious problem emerged. The author must look again at the revised manuscript and continue to revise it.

1. CFA language has been added to the abstract, and its full name should be explained when it first appears.

2. The topic is not appropriate. From the abstract, the author conducted a questionnaire and then verified the Vietnamese version of the MSQ scale. Then a new model is proposed, and the applicability of the new model is demonstrated. The core should be based on the analysis of questionnaires to build a new model.

3. A specific province has been added to the title. This province should be introduced in the introduction as a case study. Second, why do studies in this province? What is the position relationship between it and Vietnam? The method part should first clarify this.

Reviewer 3 Report

Comments and Suggestions for Authors

Accepted as it is.

Comments on the Quality of English Language

Need minor improvement 
